# Small symmetry-breaking triggering large chiroptical responses of Ag$_{70}$ nanoclusters

Xi-Ming Luo [1,2,5], Chun-Hua Gong [1,5], Fangfang Pan [3], Yubing Si [1], Jia-Wang Yuan [1,2], Muhammad Asad [1], Xi-Yan Dong [1,2✉], Shuang-Quan Zang [1✉] & Thomas C. W. Mak [1,4]

The origins of the chiroptical activities of inorganic nanostructures have perplexed scientists, and deracemization of high-nuclearity metal nanoclusters (NCs) remains challenging. Here, we report a single-crystal structure of **Rac-Ag$_{70}$** that contains enantiomeric pairs of 70-nuclearity silver clusters with 20 free valence electrons (**Ag$_{70}$**), and each of these clusters is a doubly truncated tetrahedron with pseudo-$T$ symmetry. A deracemization method using a chiral metal precursor not only stabilizes **Ag$_{70}$** in solution but also enables monitoring of the gradual enlargement of the electronic circular dichroism (CD) responses and anisotropy factor $g_{abs}$. The chiral crystals of **R/S-Ag$_{70}$** in space group $P2_1$ containing a pseudo-$T$-symmetric enantiomeric NC show significant kernel-based and shell-based CD responses. The small symmetry breaking of $T_d$ symmetry arising from local distortion of Ag−S motifs and rotation of the apical Ag$_3$ trigons results in large chiroptical responses. This work opens an avenue to construct chiral medium/large-sized NCs and nanoparticles, which are promising for asymmetric catalysis, nonlinear optics, chiral sensing, and biomedicine.

[1] College of Chemistry, Zhengzhou University, 450001 Zhengzhou, China. [2] College of Chemistry and Chemical Engineering, Henan Polytechnic University, 454003 Jiaozuo, China. [3] College of Chemistry Central China Normal University, Luoyu Road 152, 430079 Wuhan, China. [4] Department of Chemistry, The Chinese University of Hong Kong, Shatin, New Territories, Hong Kong SAR, China. [5] These authors contributed equally: Xi-Ming Luo, Chun-Hua Gong. ✉email: dongxiyan0720@hpu.edu.cn; zangsqzg@zzu.edu.cn

Chirality, an essential attribute in nature, is especially fascinating at nanosize level[1], including metal nanoclusters[2–16], quantum dots[17], one-dimensional spiral[18,19], and two-dimensional surface[20]. Chiral metal nanoclusters with atomically precise structures are the ideal models for understanding the origin of chirality at nanoscale[3]. Thus far, ultrasmall homochiral metal nanoclusters containing *ca.* less than 50 metal atoms have been more easily prepared and separated due to their rigid metal skeletons[2,5–16]. Well-defined optically pure high-nuclearity metal clusters remain scarce because of the difficulty in separating and crystallizing enantiomers. Although a few chiral medium-sized nanoclusters have been structurally resolved by single-crystal X-ray diffraction analysis[3,21–23], they crystallized in racemates, which have pairs of enantiomers in a unit cell of the crystal. Only one case of enantiomerically pure single crystals of a medium-sized (>50 metal atoms) silver nanocluster, $Ag_{78}$, protected by a chiral diphosphine ligand, has been characterized[16], yet its deracemization process is unclear, and the CD origins are not well assigned.

Herein, we report the preparation and structural characterization, including X-ray crystallography, of $\{NH_2(CH_3)_2\}_2$ $\{Ag_{70}S_4(S^iPr)_{24}(CF_3COO)_{20}(DMF)_3\}\cdot 4DMF$ (**Rac-Ag$_{70}$**). It crystallizes in the $P\bar{3}c1$ space group (No. 165) and contains enantiomeric pairs of 70-nuclearity silver clusters, each of which is a pseudo-$T$-symmetric doubly truncated tetrahedron (Fig. 1), lacking a $C_3$ axis (and thus strictly speaking should be called $C_1$ symmetry). By using a small Ag(I) cluster as a stabilizer, we obtain regularly $T_d$-symmetric $Ag_{70}$ (Fig. 1a) in the structure of achiral cocrystal **Ag$_{70}$·Ag$_{12}$**. By controlling the chiral metal precursor, we monitor a gradual increase in CD responses and $g_{abs}$ in solution, demonstrating that the chiral carboxylate enters the coordination layer step-by-step and causes a progressive deracemization of the $Ag_{70}$ racemates. Interestingly, the process can be reversed. Moreover, the chiral crystals of **R/S-Ag$_{70}$** in space group $P2_1$ only contain pseudo-$T$-symmetric enantiomeric nanoclusters similar to but with more deviation from those in **Rac-Ag$_{70}$** (Fig. 1b). The CD responses of crystalline **R/S-Ag$_{70}$** are consistent with those in solution, which originate from kernel-based and shell-based electronic transitions, as evidenced by theoretical calculations. Structural analysis reveals that the small symmetry breaking of $T_d$ symmetry arises from local distortion of Ag–S motifs and rotation of the apical $Ag_3$ trigons, resulting in large chiroptical responses.

This work opens an avenue to construct chiral medium-sized or large-sized nanoclusters and nanoparticles.

## Results

**Synthesis and characterization**. The synthesis of **Rac-Ag$_{70}$** was achieved by an acid reduction method. $\{Ag(S^iPr)\}_n$ and $CF_3COOAg$ together with $CF_3COOH$, an indispensable acid, reacted in a mixed solvent containing N,N′-dimethylformamide (DMF) and isopropanol ($^iPrOH$) under solvothermal conditions at 80 °C for 24 h. The reaction solution changed from colourless to black-red, indicating that slow reduction ($Ag^I$ to $Ag^0$) had occurred during the solvothermal process. Subsequently, black crystals of **Rac-Ag$_{70}$** were deposited after the filtrate evaporated in the dark for 1 week (Supplementary Fig. 1).

According to previous reports, $Ag^+$ ions can be reduced to $Ag^0$ atoms by heating DMF solution in a neutral or alkaline environment[24–29]. However, the silver clusters prepared under these conditions bear a maximum of two free electrons[24–27]. Here, we changed the environment from basic to acidic ($CF_3COOH$) for the preparation of **Rac-Ag$_{70}$** nanoclusters (NCs) under solvothermal conditions. At high temperature, the addition of more $CF_3COOH$ may slow down the DMF reduction and the nucleation of Ag nanoparticles, and the $S^{2-}$ anions slowly generated in situ simultaneously capture an appropriate aggregation state, resulting in a high-nuclearity structure with a distinct Ag core. Thus, a reduction method was established for the synthesis of Ag NCs possessing multiple free electrons.

A series of characterizations (detailed discussions in the Methods section and Supplementary Figs. 2–6 in Supplementary Information), including single-crystal X-ray diffraction (SCXRD), elemental analyses, powder X-ray diffraction (PXRD), thermogravimetric (TG) analyses, X-ray photoelectron spectroscopy (XPS), energy dispersive spectroscopy (EDS), elemental mapping, UV-Vis-NIR spectroscopy, and electrospray ionization mass spectrometry (ESI-MS), were applied to elucidate the structures, compositions, phase purities, and electronic states.

**Single-crystal structure of Rac-Ag$_{70}$**. Based on the SCXRD structural analysis (detailed below), 70-nuclearity Ag NCs of **Rac-Ag$_{70}$** were determined to be −2 valence-state anionic clusters with

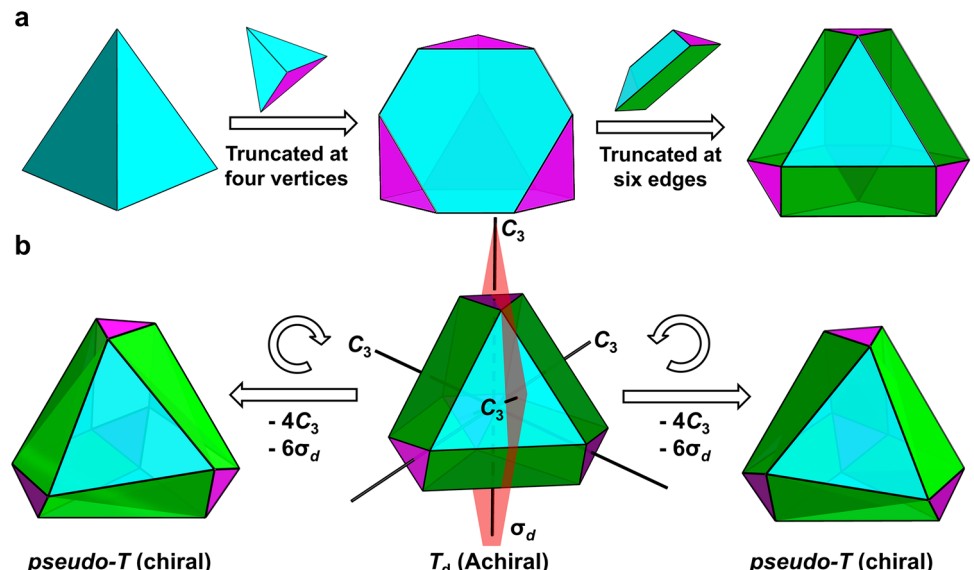

**Fig. 1 Schematic evolution of the $T_d$-symmetric doubly truncated tetrahedron and its symmetry breaking. a** Tetrahedron (one of the Platonic solids, left); truncated tetrahedron (one of the Archimedean solids, middle); doubly truncated tetrahedron (right). **b** Symmetry breaking of the $T_d$-symmetric doubly truncated tetrahedron by rotation and twisting of the apical trigons, leading to pseudo-$T$ symmetry lacking mirror planes ($\sigma_d$) and rotation axes ($C_3$).

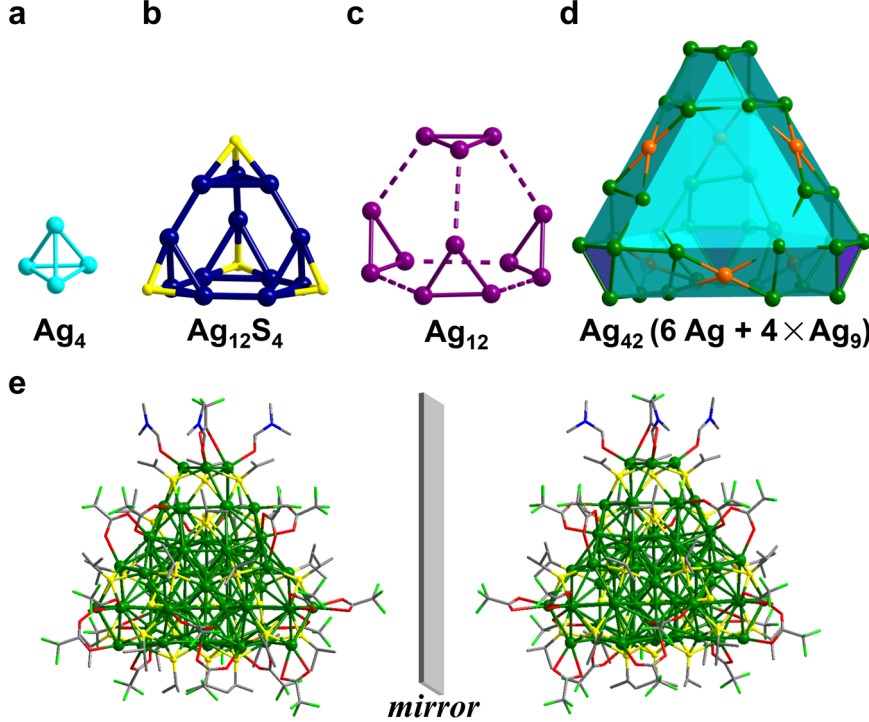

**Fig. 2 Single-crystal X-ray structure of Rac-Ag₇₀.** **a–d** Innermost $Ag_4$ tetrahedron; first-shell $Ag_{12}S_4$ tetrahedron; second-shell truncated $Ag_{12}$ tetrahedron; outermost-shell doubly truncated $Ag_{42}$ tetrahedron. **e** Enantiomers in a unit cell of **Rac-Ag₇₀**. Atom colour codes: turquoise/dark blue/violet/green/orange, Ag; yellow, S; red, O; bright green, F; blue, N; grey, C. All hydrogen atoms are omitted for clarity.

20 free valence electrons $[20 = 70(Ag^+) - 4 \times 2(S^{2-}) - 24(S^iPr^-) - 20(CF_3COO^-) + 2]$, which can be described using the jellium model $(1S^21P^61D^{10}2S^2)$[30–32]. The valence number was further corroborated by ESI-MS experiments and the well-consistent electronic transition spectra between the experimental and theoretically calculated results (vide infra).

SCXRD analysis reveals that **Rac-Ag₇₀** possesses a fascinating core-shell skeleton with a $Ag_{28}S_4$ inner kernel ($Ag_4@Ag_{12}$-$S_4@Ag_{12}$, Fig. 2a–c) enveloped by a larger doubly truncated tetrahedral $Ag_{42}$ shell (Fig. 2d) that is covered and stabilized by $S^iPr^-$ and $CF_3COO^-$ ligands together with DMF solvent molecules. The 70-Ag atoms can be divided into concentric tetrahedral multishells (Fig. 2a–d): $Ag_4$ (core)$@Ag_{12}S_4$ (1st shell) $@Ag_{12}$ (2nd shell)$@Ag_{42}$ (3rd shell). The Ag···Ag distances in **Rac-Ag₇₀** range from 2.770(2) to 3.385(2) Å, indicating the intermetallic interactions (Supplementary Figs. 7–8 and Supplementary Table 1). The detailed silver coordination modes and local structural features are discussed in the Supplementary Information (Supplementary Figs. 7–19).

From the innermost core outward (Supplementary Fig. 13), the 70-Ag atom NC, denoted $Ag_4@Ag_{12}@Ag_{12}@Ag_6@Ag_{24}@Ag_{12}$, features a tetrahedron of four Ag atoms (an idealized Platonic solid), a truncated tetrahedron of twelve Ag atoms (an idealized Archimedean solid), a truncated tetrahedron of 12 Ag atoms (an Archimedean solid), an octahedron of six Ag atoms (an idealized Platonic solid), a truncated octahedron of 24 Ag atoms (an Archimedean solid), and a doubly truncated tetrahedron of twelve Ag atoms, with each of the vertices occupied by a Ag atom. This 70-Ag atom aggregate possesses tetrahedral topology[12,21,22,33–35], which incorporates $S^{2-}$-passivated FCC-based $Ag_{16}$ inner core ($Ag-\mu_3-S$ bond lengths: 2.469(5)−2.489(6) Å; Supplementary Fig. 8). This kernel may provide a valuable clue for a deep insight into the nucleation and evolution of FCC-based Ag NCs and silver bulk materials. From single ion to three-layer

FCC close-packing ($Ag^+ \rightarrow Ag_6 \rightarrow Ag_{16}$, Supplementary Fig. 14), the regular aggregation states of small Ag species captured by four $S^{2-}$ anion-templates may represent the early-growth stage of Ag bulk or Ag nanoparticles with FCC close-packing. Based on the ideal tetrahedron growth pattern, the next larger member of this family should have a $Ag_{31}S_4$ core with FCC four-layer close-packing (Supplementary Fig. 14).

The 24 $\mu_4$-$S^iPr^-$ ligands are evenly distributed and anchored on the surface of the cluster with Ag−S bond lengths in the range of 2.432(8)−2.616(5) Å. These organic components can be divided into two groups according to tetrahedral arrangement characteristics (Supplementary Fig. 15). The 20 $CF_3COO^-$ ligands, which are located on the rectangular or triangular faces of the double truncated tetrahedron protect the $Ag_{70}$ in $\mu_2$-$\eta^1,\eta^1$ or $\mu_1$-$\eta^1,\eta^1$ coordination mode (Ag−O bond lengths: 2.29(2) −2.44(4) Å, Fig. 2e and Supplementary Fig. 13a). Three DMF molecules are located at a $Ag_3$ vertex of polyhedra (Ag−O bond lengths: 2.35(2) Å). In addition, it was found that there are multiple C − H···F hydrogen bonds between $CF_3COO^-$ and −$CH_3$ groups of $S^iPr^-$ and DMF ligands (Supplementary Fig. 16), suggesting the strong interactions between the clusters.

Careful analysis of the geometric structure of an individual cluster in **Rac-Ag₇₀** indicated that it lacks a mirror plane ($\sigma$) and inversion ($i$). Meanwhile, breakage of $T_d$ symmetry occurred due to distortion of the surface Ag−S motifs[3], although the rotation angle along the pseudo-$C_3$ axes is slight (Supplementary Fig. 17). Notably, the inner kernel ($Ag_4$ (core) and $Ag_{12}S_4$ (1st shell), Supplementary Fig. 18) presents achiral structural characteristics due to the existence of the three mirror planes. This slight change based on surface motifs could be an important cause of large chiroptical responses of inorganic nanoparticles, which will be further verified by homochiral single crystals. As a result, left- and right-handed enantiomers coexist in the unit cell of **Rac-Ag₇₀** (Fig. 2e and Supplementary Fig. 17).

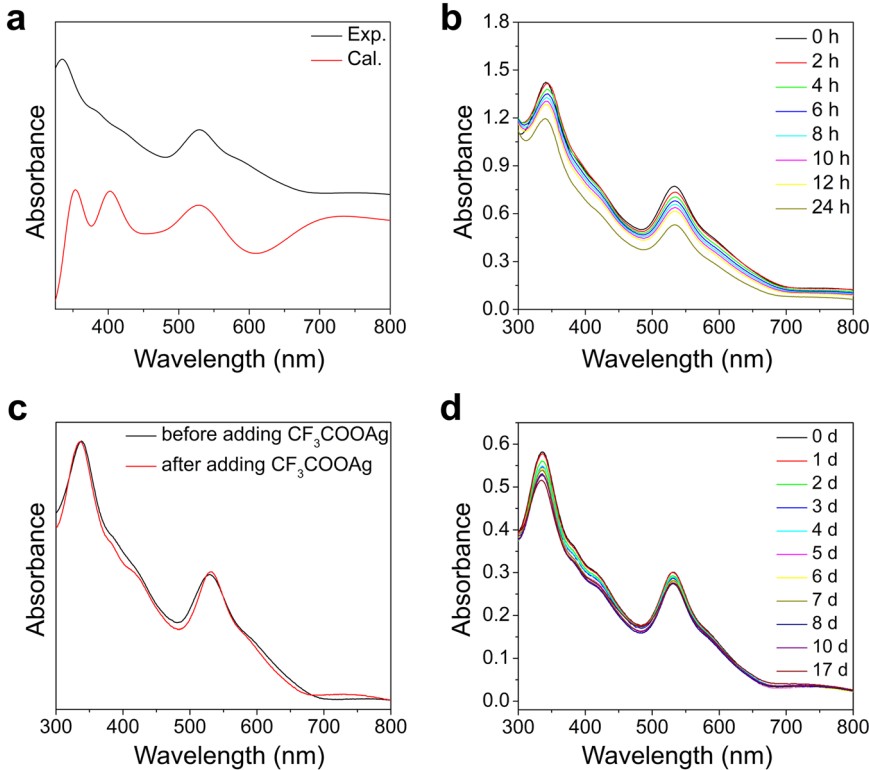

**Fig. 3 UV-Vis absorption spectra of Rac-Ag₇₀. a** Experimental (black) and calculated (red) absorption spectra of **Rac-Ag₇₀** in EtOH. **b** Time-dependent UV-Vis absorption spectra of **Rac-Ag₇₀** in EtOH (ca. 5 × 10⁻⁶ M) under ambient conditions. **c** UV-Vis absorption spectra of the EtOH solution of **Rac-Ag₇₀** before and after adding silver trifluoroacetate (250 eq.). **d** Time-dependent UV-Vis absorption spectra of **Rac-Ag₇₀** in EtOH together with silver trifluoroacetate (250 eq.).

**Optical properties and stability of Rac-Ag₇₀ in solution.** The UV-Vis-NIR absorption spectrum of the **Rac-Ag₇₀** solution exhibits multiple molecular-like peaks: two sharp peaks (335 nm, 530 nm), one broad peak (ca. 767 nm), some shoulder peaks (ca. 386 nm, 426 nm, and 595 nm) and the lowest absorption peak at ~1450 nm (0.855 eV) (Fig. 3a and Supplementary Figs. 20–21). The main peaks that appear in the UV-Vis absorption spectrum of **Rac-Ag₇₀** (Fig. 3a) are well reproduced in the theoretically simulated spectrum (the calculations are described in the Methods section and Supplementary Information), further confirming the structure and the number of valence electrons of this NC[7,13]. The efficient transitions of the low-energy absorption peaks mainly involve metal-based transitions (Supplementary Figs. 22–26), and partial ligands (CF₃COO⁻) contribute to the high-energy peaks at 353 nm (Supplementary Fig. 25). The solution of **Rac-Ag₇₀** shows broad emission peaks centred at 1300 nm in the NIR-II region (Supplementary Fig. 27), which may arise from core-based transitions with low energy[14,36].

Furthermore, ESI-MS was used to investigate the composition, charge state, and solution behaviour of **Rac-Ag₇₀** in detail (Supplementary Figs. 28–29). Crystals of **Rac-Ag₇₀** were dissolved in EtOH and measured in negative-ion mode with varied declustering potential and collision energy, showing three main grouped peaks in the mass-to-charge ratio (m/z) ranges of 3500–4000 (−3 charge state), 4200–4600 (−2 charge state), and 5500–6100 (−2 charge state). As shown in Supplementary Fig. 29, the peaks (**Rac-Ag₇₀**: 2 h − 2 l) corresponding to the complete cluster skeleton (Ag₇₀S₄) are easy to find, confirming that the entire cluster molecule can stably exist in solution. The correlations between the species mainly involve the same Ag₇₀S₄ skeleton and the exchange of molecules outside the shell (SⁱPr⁻ and CF₃COO⁻), in which the weakly bound CF₃COO⁻

molecules are easily removed from the surface of the cluster under the prevailing ionization conditions. When a collision energy is applied to the system (from 0 to −15 V), fragment peaks 1a − 1j appear, and their intensity gradually increases; these peaks may be ascribed to some large fragments (Ag₍₄₈₋₅₁₎S₄, Supplementary Fig. 29b and Supplementary Table 2), suggesting that **Rac-Ag₇₀** is unstable in solution, consistent with the results of time-dependent UV-Vis (Fig. 3b). Meanwhile, the presence of peaks 2a − 2c (Ag₆₈S₄), and 2d − 2 g (Ag₆₉S₄) indicates that the dissociation of **Rac-Ag₇₀** starts with the loss of CF₃COOAg small molecules step-by-step. The appearance of the peak corresponding to the Ag₁₂ cluster ([Ag₁₂(SⁱPr)₆(CF₃COO)₇]⁻, Supplementary Fig. 29d) indicates that this stable small cluster formed immediately after **Rac-Ag₇₀** decomposition.

Inspired by the relatively strong interactions between the anionic cluster and the small molecules (CF₃COOAg), as revealed by ESI − MS (2 m − 2t, Supplementary Fig. 29c and Supplementary Table 2), we proposed that CF₃COOAg could stabilize the **Rac-Ag₇₀** cluster and that dissociation-coordination equilibrium could occur between the anionic NC and CF₃COOAg. This speculation was confirmed by time-dependent UV-Vis absorption spectra and ESI-MS experiments. When 250 equivalents of CF₃COOAg were added to the solution of **Rac-Ag₇₀**, Ag₇₀ remained stable for dozens of days (Fig. 3c–d). As depicted in Supplementary Fig. 29h–i, supramolecular assembly was carried out in a solution of **Rac-Ag₇₀** after the introduction of 250 eq. of CF₃COOAg to form a more stable hybrid product, **Ag₇₀**·(1–4) CF₃COOAg (Supplementary Table 3), which completely inhibited the removal of CF₃COO⁻ and the dissociation of clusters. When C₂F₅COOAg was used, we obtained the same stability trend of **Rac-Ag₇₀** in solution (Supplementary Fig. 30 and Supplementary Tables 4–5). This direct and sufficient evidence suggested that

stable and rapid dissociation-coordination equilibrium occurred between the anionic NC and the small molecules (RCOOAg) in the solution. Based on the above experimental data, a reasonable stability mechanism in solution is proposed. For the solution of **Rac-Ag$_{70}$**, NCs decompose and form small molecules (such as CF$_3$COOAg) and fragments (such as Ag$_{12}$ clusters) until equilibrium is reached. Note that when the concentration of **Rac-Ag$_{70}$** is too low, the NCs will be completely destroyed. If there is a large number of small molecules in the **Rac-Ag$_{70}$** solution, then the decomposition of the NCs will be inhibited, and some stable products (**Ag$_{70}$·(1–4)CF$_3$COOAg**) will be formed. Note that excessive RCOOAg (more than 500 eq.) is detrimental to these NCs.

**Deracemization of Rac-Ag$_{70}$ solution.** Considering that each cluster in **Rac-Ag$_{70}$** is slightly distorted and thereafter chiral and that RCOOAg not only stabilized the NC in solution but also replaced the auxiliary CF$_3$COO$^-$ ligands on the constant metal framework of Ag$_{70}$, chiral trifluorolactic acid (HTFL) was used to deracemize **Rac-Ag$_{70}$**. The as-prepared solution of **Rac-Ag$_{70}$** was treated with different amounts of silver R-/S-trifluorolactate (denoted R-/S-TFLAg) in EtOH. The resulting mixture was monitored by UV-Vis, CD, ESI–MS and $^{19}$F-NMR spectra (Fig. 4 and Supplementary Figs. 31–40). As shown in Fig. 4a and Supplementary Figs. 31–32, the as-prepared solution of **Rac-Ag$_{70}$** is CD silent. With the continuous introduction of R-/S-TFLAg, Cotton effects of the solution appear (with weak peaks at 320, 530, 600, and 700 nm (broad) and strong peaks at 280, 383, 420, and 469 nm) (Fig. 4a, d). The Cotton effects and dissymmetry factor $g_{abs}$ (Supplementary Fig. 33) gradually increase with an increasing ratio of n(R-/S-TFLAg):n(**Rac-Ag$_{70}$**), while the UV-Vis absorption behaviour is basically unchanged (Supplementary Figs. 31–32). Note that R-/S-TFLAg in EtOH exhibits only CD optical activity below 250 nm (Supplementary Fig. 34).

The corresponding ESI-MS spectrum (Supplementary Fig. 35 and Supplementary Tables 6–7) shows the grouped peaks assigned to {Ag$_{70}$S$_4$(S$^i$Pr)$_{24}$(TFL/CF$_3$COO)$_{21}$}$^{3-}$·(0–3)TFLAg and {Ag$_{70}$S$_4$(S$^i$Pr)$_{24}$(TFL/CF$_3$COO)$_{20}$}$^{2-}$·(0–4)TFLAg, suggesting that the increase of Cotton effects is positively associated with gradual ligand exchange (CF$_3$COO$^- \rightarrow$TFL$^-$). In addition, the corresponding UV-Vis absorption does not change within a certain period of time (Supplementary Fig. 31d), indicating that the main metal skeleton of the cluster remains unchanged, and dynamic equilibrium and long-term stability of the system are achieved. Interestingly, in the range of n(R-/S-TFLAg):n(**Rac-Ag$_{70}$**) = 0–15, the Cotton effects are almost linearly enhanced, while beyond the ratio of 15, these effects are basically unchanged (Fig. 4c and Supplementary Fig. 32c). The amplification of chiral signals may stem from the fact that the achiral ligands (CF$_3$COO$^-$) are continuously replaced by chiral ligands (TFL$^-$) on the cluster surface, and the symmetry breaking of the whole structure is gradually amplified. A reversible decrease in Cotton effects is recovered by using CF$_3$COOAg titration (Fig. 4b). In addition, the reversible CD titration does not change the nature of the cluster (Supplementary Figs. 31–32, Supplementary Figs. 36–37 and Supplementary Tables 8–9), which could find applications in chiral sensing fields.

**Cocrystals of Ag$_{70}$·Ag$_{12}$ evidencing the proposed stability mechanism.** Aiming to further verify the above-proposed stability mechanism in solution, we attempted to prepare single crystals that contained an anionic Ag$_{70}$ cluster and a small Ag(I)$_{12}$ cluster ({Ag$_{12}$(S$^i$Pr)$_6$(CF$_3$COO)$_6$}) as a stabilizer, which was named **Ag$_{70}$·Ag$_{12}$** (Supplementary Figs. 41–57)[37–43]. The appearance of Ag(I)$_{12}$ cluster could be attributed to the remainder of the unreduced silver(I) fragments in the EtOH-DMF system. The smaller Ag(I)$_{12}$ cluster possesses a slightly distorted cuboctahedral metal framework (Ag···Ag distances: 3.06(1)–3.09(2) Å,

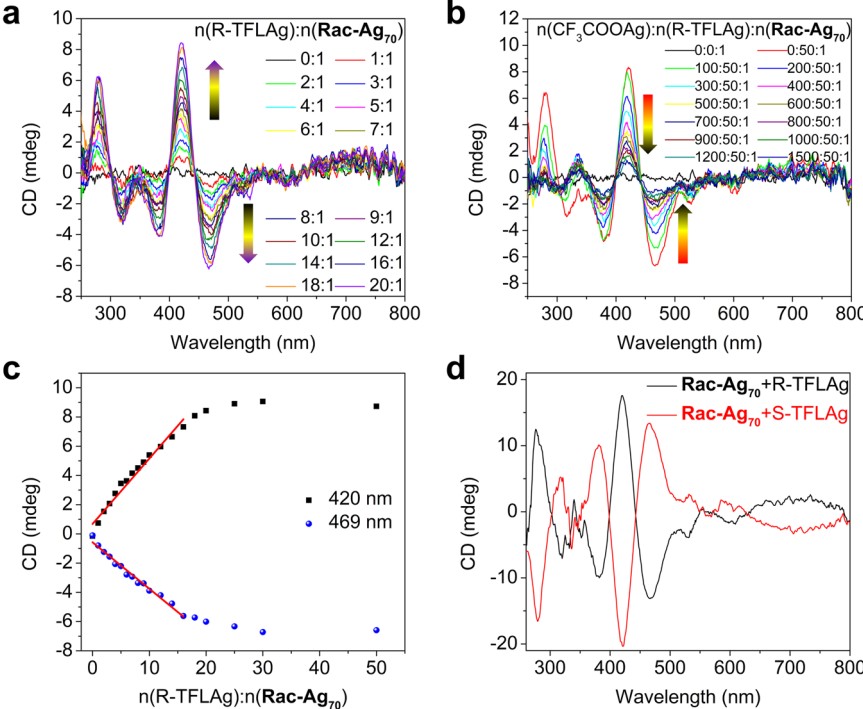

**Fig. 4 Deracemization and Cotton effect amplification of Rac-Ag$_{70}$ in solution. a** CD titration of **Rac-Ag$_{70}$** using a chiral metal precursor (R-TFLAg) in EtOH. **b** Back CD titration of deracemization solution using an achiral metal precursor (CF$_3$COOAg) in EtOH. **c** Cotton effects at 420 nm and 469 nm plotted vs. the concentration ratio of R-TFLAg and **Rac-Ag$_{70}$**. **d** CD spectra of deracemization of **Rac-Ag$_{70}$** solution.

Supplementary Figs. 47–49). The six $\mu_4$-S$^i$Pr$^-$ ligands are anchored on the quadrilateral surface of Ag(I)$_{12}$ with Ag−S bond lengths in the range of 2.44(4)–2.50(4) Å, and the CF$_3$COO$^-$ ligands on the edges in $\mu_2$-$\eta^1$, $\eta^1$ coordination mode (Ag−O bond lengths: 2.29(2)–2.44(4) Å).

Cocrystallization with Ag(I)$_{12}$ leads to a more regular stacking of Ag$_{70}$ (Supplementary Figs. 50–51); thus, **Ag$_{70}$·Ag$_{12}$** crystallizes in a higher-symmetry space group ($Fd\bar{3}m$, number 227) compared to **Rac-Ag$_{70}$** ($P\bar{3}c1$, number 165). More interestingly, the 70-nuclearity Ag NC in **Ag$_{70}$·Ag$_{12}$** exhibits nearly perfect $T_d$ symmetry (Supplementary Fig. 47b), and its polyhedral skeleton is similar to that in **Rac-Ag$_{70}$**, except that the slight symmetry breaking leads to different Ag···Ag distances (Supplementary Figs. 48–49 and Supplementary Table 1). Therefore, the small Ag(I)$_{12}$ cluster not only stabilizes **Ag$_{70}$** but also changes the crystal packing, which probably weakens the intercluster strain in **Rac-Ag$_{70}$** and results in $T_d$-symmetric Ag$_{70}$ in **Ag$_{70}$·Ag$_{12}$**.

**Ag$_{70}$·Ag$_{12}$** exhibits UV-Vis-NIR absorption (Supplementary Figs. 52–53), NIR-II photoluminescence emission (Supplementary Fig. 27) and ESI-MS results (Supplementary Figs. 54–55 and Supplementary Table 10) similar to those of **Rac-Ag$_{70}$**. The Ag(I)$_{12}$ cluster truly stabilizes Ag$_{70}$ in solution: **Ag$_{70}$·Ag$_{12}$** remains stable in EtOH for more than 7 days, as revealed by time-dependent UV-Vis absorption spectra (Supplementary Fig. 56); **Ag$_{70}$·Ag$_{12}$** is also more thermally stable than **Rac-Ag$_{70}$** in solution (Supplementary Fig. 57).

**Single crystal of chiral Ag$_{70}$ verifying the proposed deracemization mechanism.** We subsequently attempted to prepare single crystals of homochiral Ag$_{70}$. This is considerably difficult because homochiral high-nuclearity NCs remain an open challenge[9,13,16]. After continuous attempts, we obtained enantiomeric **R/S-Ag$_{70}$** crystals (Supplementary Fig. 58) in chiral space group $P2_1$ in the presence of R/S-TFLAg. The main composition of **R/S-Ag$_{70}$** was determined as {Ag$_{70}$S$_4$(S$^i$Pr)$_{24}$(CF$_3$COO)$_{12}$(R/S-TFL)$_8$}$^{2-}$ based on SCXRD, ESI-MS (Supplementary Figs. 35–37) and $^{19}$F-NMR (Supplementary Fig. 59). Although the peripheral ligands of **R-Ag$_{70}$** could not be positioned, its Ag−S skeleton was well resolved (see the Methods for the refinement of the structure of **R-Ag$_{70}$**). Structural analysis indicates that the pseudo-$T$-symmetric metal skeleton of **R-Ag$_{70}$** (Fig. 5a–b) is more severely distorted than that of **Rac-Ag$_{70}$** (Supplementary Figs. 60–63 and Supplementary Table 1). Therefore, the absence of mirror planes in the Ag$_{70}$ cluster molecule leads to the obvious shell-based and metal kernel-based Cotton effects of **R-Ag$_{70}$** and **S-Ag$_{70}$** crystals (Fig. 5c and Supplementary Fig. 64). The CD spectra of **R-Ag$_{70}$** and **S-Ag$_{70}$** are consistent with those of the above deracemized **Rac-Ag$_{70}$** solution with R/S-TFLAg, verifying the small symmetry breaking related to the considerable Cotton effects. Combining the single-crystal structure analysis, and the nearly identical CD

signals of solution and solid state, we propose that the incoming TFL$^-$ (chiral organic ligand) induces the structural deformation, which plays a critical role in chiroptical response and the deracemization process. In addition, TFLAg (chiral complex), which was used to trigger the reaction in our experiment, is found to be the concomitants of the silver cluster in ESI-MS (Supplementary Figs. 35–37 and Supplementary Tables 6–9), and also plays a role in stabilizing the clusters (Supplementary Fig. 31d and Supplementary Fig. 40).

## Discussion

In summary, we prepared a medium-sized silver NC racemate (**Rac-Ag$_{70}$**) through acid reduction synthesis, in which each NC featured the largest doubly truncated tetrahedron with pseudo-$T$ symmetry. The dissociation-coordination mechanism between ligands and the silver framework enables stabilization of Ag$_{70}$ by cocrystallization with Ag(I)$_{12}$ clusters, deracemization of **Rac-Ag$_{70}$** in solution, and achievement of enantiomeric crystals of **R/S-Ag$_{70}$** through chiral metal precursors. SCXRD analysis revealed that small symmetry breaking from $T_d$ symmetry is responsible for the large chiroptical response of chiral clusters. This work provides not only an insight into the stabilization mechanism of high-nuclearity metal clusters and symmetry breaking related to the chiroptical response but also a significant case for the exploration of growth of truncated tetrahedron-shaped noble-metal NCs.

## Methods

**Synthesis of Rac-Ag$_{70}$.** {Ag(S$^i$Pr)}$_n$ (0.05 mmol, 9 mg) and CF$_3$COOAg (0.10 mmol, 22.1 mg) were dissolved together in a mixed solvent of $^i$PrOH and DMF (5.0 mL, $v$:$v$ = 7:3), and then, 50 μL of CF$_3$COOH was added to the above solution. The mixture was sealed in a 15 mL Teflon-lined reaction vessel and kept at 80 °C for 24 h. After cooling to room temperature, the black-red solution was filtered and evaporated in the dark for 1 week. Black needle crystals of **Rac-Ag$_{70}$** were isolated and washed with dichloride/n-hexane at a yield of 35% (based on {Ag(S$^i$Pr)}$_n$). Elemental analysis (found (calcd), %; based on C$_{137}$H$_{233}$O$_{47}$Ag$_{70}$N$_9$S$_{28}$F$_{60}$): C, 13.67 (13.33); H, 1.94 (1.90); N, 0.85 (1.02).

**Synthesis of Ag$_{70}$·Ag$_{12}$.** {Ag(S$^i$Pr)}$_n$ (0.05 mmol, 9 mg) and CF$_3$COOAg (0.10 mmol, 22.1 mg) were dissolved together in a mixed solvent of EtOH and DMF (5.0 mL, $v$:$v$ = 7:3), and then, 50 μL of CF$_3$COOH was added to the above solution. The mixture was sealed in a 15 mL Teflon-lined reaction vessel and kept at 80 °C for 24 h. After cooling to room temperature, the black-red solution was filtered and evaporated in the dark for 1 week. Black octahedral block crystals of **Ag$_{70}$·Ag$_{12}$** ({NH$_2$(CH$_3$)$_2$}$_2${Ag$_{70}$S$_4$(S$^i$Pr)$_{24}$(CF$_3$COO)$_{20}$(DMF)$_x$}·{Ag$_{12}$(S$^i$Pr)$_6$ (CF$_3$COO)$_6$}·(DMF)$_{12-x}$) were isolated and washed with dichloride/n-hexane at a yield of 20% (based on {Ag(S$^i$Pr)}$_n$). Elemental analysis (found (calcd), %; based on C$_{188}$H$_{324}$O$_{66}$Ag$_{82}$N$_{16}$S$_{34}$F$_{78}$): C, 14.49 (14.44); H, 1.99 (2.06); N, 0.98 (1.29).

**Synthesis of R/S-Ag$_{70}$.** Chiral **R/S-Ag$_{70}$** was synthesized by a ligand-exchange method from **Ag$_{70}$·Ag$_{12}$** NCs. Herein, the synthetic method for {Ag$_{70}$S$_4$(- S$^i$Pr)$_{24}$(CF$_3$COO)$_{20-n}$(R-TFL)$_n$}$^{2-}$ (**R-Ag$_{70}$**, $n$ = 8 based on $^{19}$F-NMR (Supplementary Fig. 59)) is used as an example. Three milligrams of **Ag$_{70}$·Ag$_{12}$** were dissolved in 2 mL DMF. One milligram of R-TFLAg was added to the above

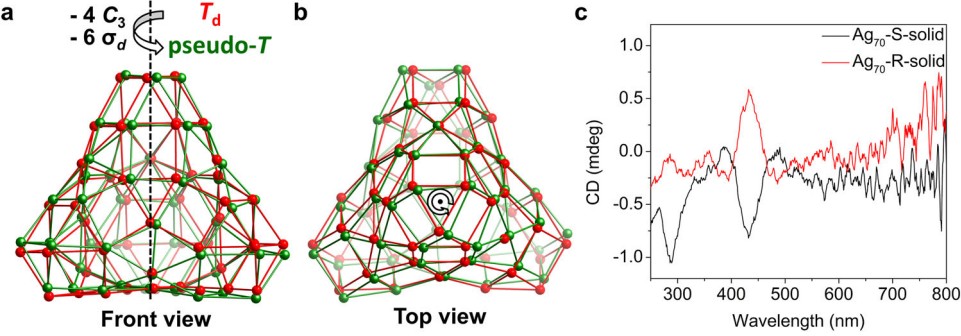

**Fig. 5 Symmetry breaking of the $T_d$-symmetric shell of Ag$_{70}$ and solid-state CD spectra. a, b** Compared with the $T_d$ M$_{54}$ doubly truncated tetrahedron (red), an illustration of the symmetry breaking in the Ag$_{54}$ shell (green) of **R-Ag$_{70}$**. **c** CD spectra of **R/S-Ag$_{70}$** crystallites.

solution under stirring and stirred for 2 min. The black-red solution was filtered and evaporated in the dark for 1 week. The black block crystals of **R-Ag₇₀** were isolated and washed with dichloride/n-hexane (yield: 70 %).

**Crystallographic data collection and refinement of the structure**. SCXRD measurements of **Rac-Ag₇₀**, **Ag₇₀·Ag₁₂** and **R-Ag₇₀** were performed at 200 K on a Rigaku XtaLAB Pro diffractometer with Cu-K$\alpha$ radiation ($\lambda$ = 1.54184 Å). Data collection and reduction were performed using the program CrysAlisPro[44,45]. All structures were solved with direct methods (SHELXS)[46] and refined by full-matrix least squares on $F^2$ using OLEX2[47], which utilizes the SHELXL-2018/3 module[48]. All non-hydrogen atoms were refined anisotropically, and hydrogen atoms were placed in calculated positions with idealized geometries and assigned fixed isotropic displacement parameters. Appropriate restraints and/or constraints were applied to the geometry, and the atomic displacement parameters of the atoms in the clusters were determined. The absence of {NH₂(CH₃)₂}⁺, CF₃COO⁻, and DMF molecules in the SCXRD data of **Ag₇₀·Ag₁₂** could be caused by weak diffraction, high symmetry, and a highly disordered state in the lattice.

Due to the extremely large unit cell and high disorder on the ligands, it is not very easy to solve the structure of the whole cluster for **R-Ag₇₀**, particularly for the organic ligands. Even, to be honest the space group cannot be confidently confirmed from the X-ray diffraction data at 100 K. According to the results from XPREP, other than the suggested "A" lattice by default, primitive lattice should be the more suitable one since there are in total 15139 exceptions of $I > 3\sigma$ and the mean intensity is 10.2 with the man $I/\sigma$ of 1.9. The exceptions for other lattices show similar total reflection number, and only slightly high mean intensity (21.4–27.7) and mean $I/\sigma$ (2.4–2.6). The high statistic mean |E*E−1| value (1.019 in this case) normally points to the centrosymmetry. However, a large number of reflection exceptions tend to suggest the absence of any glide planes (a, n, c) in the structure. In detail, comparing with that only 30 weak reflections ($I_{mean}$ = 1.1, $I/\sigma$ = 0.5) violate the 2₁ axis, 1835 reflections where 86 have intensity stronger than $3\sigma$ don't support the existence of c glide. The exceptions for a and n symmetries show similar total number as that for c (1835 for a and 1843 for n) (Supplementary Fig. 65). Therefore, the possible space group might be P2₁. The confusing part is that much more strong ($I/3\sigma$) reflections violate the a and c plane, which suggests the P2₁/c as an alternative. Considering the ligand that we used to synthesize the cluster is chiral with analytical pure (97%) and the relatively high yield (70%) of the single-crystal product, the correct space group of **R-Ag₇₀** should be P2₁. Finally, we solved the structure in P2₁ space group. Due to the bad disorder on the organic ligands, only the Ag₇₀S₂₈ core can be assigned and freely refined with anisotropic displacement parameters. With this incomplete model, a Flack parameter of 0.42(6) was obtained, indicating the twinning. However, the apparently high Flack parameter is also possibly caused by the lack of the organic part of the structure, particularly the chiral ligand.

Similar cell parameters ($a$ = 37.478(3) Å, $b$ = 33.6617(11) Å, $c$ = 23.8119(11) Å, $\alpha$ = 90°, $\beta$ = 90.938(6)°, $\gamma$ = 90°) were also collected for **S-Ag₇₀** through the SCXRD. Unfortunately, we didn't obtain the single crystals of **S-Ag₇₀** with enough quality.

Detailed information about the X-ray crystal data, intensity collection procedure, and refinement results for **Rac-Ag₇₀**, **Ag₇₀·Ag₁₂** and **R-Ag₇₀** are summarized in Supplementary Table 11.

**Quantum chemical calculations**. The ground state of the metal cluster for Ag₇₀ was optimized by the semiempirical tight binding method (GFN-xTB) with the GBSA model for methanol[49], and all excitations up to 6 eV were calculated with the simplified Tamm-Dancoff approach (sTDA)[50–53]. The molecular orbitals were extracted from Molden format by Multiwfn[54] and then rendered and virtualized by the VMD program[55].

## Data availability

Data supporting the findings of this manuscript are available from the corresponding authors upon request. The X-ray crystallographic coordinates for structures reported in this article have been deposited at the Cambridge Crystallographic Data Centre (CCDC) under deposition numbers CCDC: 2072308 (**Rac-Ag₇₀**), 2072309 (**Ag₇₀·Ag₁₂**), and 2104598 (**R-Ag₇₀**).

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

## Acknowledgements

This work was supported by the National Natural Science Foundation of China (No. 92061201, 21825106, U21A20277, and 21975065) and Zhengzhou University. We thank Prof. Stefan Grimme and Dr. Marc de Wergifosse for their helpful comments on the part of computational methods, and thank for the support of parallel high performance computing of National Supercomputing Center in Zhengzhou.

## Author contributions

S.Q.Z. conceived and designed the experiments. X.M.L., C.H.G., and J.W.Y. conducted the synthesis and characterization. X.M.L. drew pictures in the manuscript. S.Q.Z., X.Y.D., and X.M.L. analyzed the experimental results. X.M.L. and F.P. completed crystallographic data collection and refinement of the structure. Y.S. performed the calculations. M.A. helped to revise the writings. X.M.L., X.Y.D., S.Q.Z., and T.C.W.M. co-wrote the manuscript.

## Competing interests

The authors declare no competing interests.
