## [Peer Review File · Nature Communications]

Small Symmetry-Breaking Triggering Large Chiroptical Responses of Ag70 NanoclustersREVIEWER COMMENTS

Reviewer #1 (Remarks to the Author):

Review of NCOMMS-21-33371

This is an excellent and well-written paper on a new Ag₇₀ nanocluster." It deserves serious consideration for publication in Nature Communications. However, in its present form, the manuscript has several "deficiencies" that need to be ratified:

1. The formulation of the title cluster is $\{Ag_{70}S_4(S\text{-iPr})_{24}(TFA)_{20}(DMF)_3\}^{2-}$ (Rac-Ag₇₀). Here CF₃COO⁻ is abbreviated as TFA. What is the formulation of companion cluster Ag₁₂ in the cocrystal Ag₇₀:Ag₁₂? It looks like this small Ag₁₂ cluster is a highly distorted cuboctahedron (Fig. S43). Yet the large Ag₇₀ cluster, under the influence of this small companion Ag₁₂ cluster, has "reverted" back to achiral of T_d symmetry! With TFA ligands, Ag₇₀ crystallizes as a racemate Rac-Ag₇₀ with mirror-images of R and S chiral clusters: how did this small, distorted Ag₁₂ cluster, help make the large Ag₇₀ cluster lose its chirality?
2. Please provide structural details of the Ag₁₂ cluster, complete with the ligand positions and interatomic distances, in the cocrystal Ag₇₀:Ag₁₂. Also provide the UV-Vis and CD of the Ag₁₂ cluster in pristine form if possible. If this Ag₁₂ cluster cannot be made, I would like to suggest DFT calculations to identify its contribution to the UV-Vis and CD spectra of the cocrystal, if any.
3. As the TFA is replaced by the chiral TFL ligands, the CD of the Ag₇₀ is progressively enhanced, indicating deracemization. Is this experimental observation the result of (1) the instant conversion of, say, S to R enantiomer, or (2) a progressive distortion of the metal framework as more and more TFAs are replaced by the chiral TFLs? If the answer is (1), how many TFAs must be replaced by chiral TFLs to cause this transformation?
4. The UV-Vis remains unchanged throughout the TFA-TFL exchange reaction. More importantly, this exchange reaction should be followed/tracked (in sync) by CD, FTIR, NMR to determine the ratio of TFL:TFA ligands on the surface. This ratio may or may not (my guess is NOT) correspond to the experimental ratio used. The purpose of this tracking is to determine the m value in $\{Ag_{70}S_4(S\text{-iPr})_{24}(TFL)_{20-m}(TFA)_m(DMF)_3\}^{2-}$ and to correlate it with the CD signal.
5. Fig. 32c provides ESI-MS of $\{Ag_{70}S_4(S\text{-iPr})_{24}(TFL)_{20-m}(TFA)_m(DMF)_3\}^{2-x}(TFL)Ag$. What is the role of the extra (TFL)Ag complexes? Are they surface attachments?
6. This paper provides no structural details on the surface ligands for the STAR cluster: the chiral R-Ag₇₀. What is the TFL:TFA ratio on the surface? Are they badly disordered? Perhaps FTIR or NMR can provide an estimate (see Item 4

above).

7. It was stated in the paper that “Only one case of enantiomerically pure single crystals of a medium-sized silver nanocluster, Ag₇₈, protected by a chiral diphosphine ligand, has been characterized.²¹” This is utterly not true. The following are just a few prior examples:

Angew. Chem. Int. Ed., **2021**, DOI: 10.1002/anie.202108141; *Angew. Chem. Int. Ed.*, **2021**, *60*(23): 12897-12903; *J. Am. Chem. Soc.*, **2021**, *143*(27): 10214-10220; *Angew. Chem. Int. Ed.* **2018**, *57*(13): 3421-3425.

Reviewer #2 (Remarks to the Author):

Crystallography Report

This took me a while because I am entering unfamiliar territory here. For this study, the question is not: Are these fully refined crystal structures? They clearly aren't, and don't claim to be. The question is rather: Do the data support the conclusions the authors draw from them? My tentative answer to this is: As far as I can tell they do. I have my doubts about the charges of the clusters, simply due to the use of SQUEEZE, but from what I can tell the main conclusions about the chiral shape of the clusters are not affected by the number of valence electrons within it (and I do not understand the spectroscopic evidence, so whether that supports the charges of the clusters other referees have to answer).

I have a slight reservation about the P21 structure: I agree with the authors that the most likely space group is indeed P21; I have tried to solve the structure in P21/c and got nowhere, and ADDSYM from PLATON did not suggest any higher symmetry. However, as the authors state correctly the FLACK parameter is very high, and this can sometimes resolve itself upon full refinement of the structure. What I have also sometimes observed is that distortions in molecules occur together with high FLACK parameters in early refinement stages which both disappear when the structure is fully resolved. I cannot rule out from the data presented here that this has not occurred here, i.e. that the small differences between the AG70 cluster in P-3c1 and in P21 are due to incomplete refinement. Again, I do not understand the spectroscopic data about the shape of the clusters, so if that indeed shows that the additional distortion is real then my comment above is void.

Reviewer #3 (Remarks to the Author):

The article can be published after the following questions are addressed.

In the structural description, only the metal framework was discussed. The coordination of thiolates, trifluoroacetates, and DMF ligands should be added. Are there any changes of Ag...Ag, Ag-S, and Ag-O bonds between T- and Td-symmetric Ag70? The bond lengths or angles should be reported, which are the benchmark to identify whether the structure is distorted.

The crystals of Rac-Ag70 and Ag70·Ag12 were obtained from the same synthetic procedure (in experimental section, page 14). How did the authors said that "By using a small Ag(I) cluster as a stabilizer,..."? Based on the description of the experimental section, those two crystals can be obtained in the same condition. How to separate the two in the same reaction since the colors of the crystals are similar?

The metal framework possesses Td symmetry in Ag70·Ag12. Will it be the same symmetry if it contains the surficial ligands? In page 12, the author mentioned that "the small Ag(I)12 cluster not only stabilizes Ag70 but also changes the crystal packing, which probably weakens the intercluster strain in Rac-Ag70 and result in Td-symmetric Ag70 in Ag70·Ag12." Are there any interactions such as H...F contacts between clusters in Rac-Ag70? If so, please consider them for discussion.

It is clever to add chiral ligands (TFL-) to get optically pure R-/S-Ag70. R-Ag70 crystallized in a non-centrosymmetric space group P21, but the flack parameter (0.42) shows large uncertainty of the determination of the absolute structure. It could be due to racemic or twinned. However, the CD spectrum shows optical activity nicely. Thus this reviewer believes the R-/S-Ag70 were synthesized. The Figure 4d is quite unclear while R-/S- TFLAg and Rac-Ag70+R/S-TFLAg are stacked together. Please revise.

Line 352, Page 15, "no-H" should be " non-hydrogen".

Line 180, Page 9, "m/z" should be "*m/z*" (in italic style)

Reviewer #4 (Remarks to the Author):

Small Symmetry-Breaking Triggering Large Chiroptical Responses of Ag₇₀Nanoclusters - Review

This paper presents the structure and optical properties of chiral Ag-70 nanoclusters. A crystal structure of racemic Ag₇₀ nanoclusters is first presented. This material was then de-racemized by incorporating a chiral silver precursor into the synthesis, and the crystals of resulting R/S enantiomeric pairs were also solved. De-racemization was also monitored using CD and was found to be reversible. In particular, deracemization of such clusters for R/S structure solutions is of particular significance, as it remains a challenge.

The charge state of rac-Ag₇₀ is determined to be -2, based on the jellium model, and supported by ESI-MS data. The UV-Vis-IR spectra also conform to that expected of a molecular cluster, and are supported by theoretical simulations.

The structural analysis provided suggests the surface Ag-S moieties cause a minor T_d symmetry breaking distortion, resulting in two enantiomers with large chiroptical properties. The CD data supports this. However, the author's note that the ligand layer structure was unable to be solved with single crystal XRD, which is not entirely unexpected given the fragility of similar metal nanoclusters when exposed to X-rays.

A co-crystal of Ag₇₀-Ag₁₂ was also solved, and found to impart greater stability than the Rac-Ag₇₀ in solution. A stability mechanism of Ag₇₀ clusters in solution is proposed. The paper also presents a novel Ag₁₆S₄ FCC inner core structure within the Ag₇₀ cluster, which is particularly of interest to the community.

On the whole, the data in the paper is convincing and well presented, and should be published for the community. I do not have major remarks other than the following:

A few parts of the main text are unclear, which could undermine the importance of this work. In particular, the introductory sentences are somewhat clunky/confusing (lines 40 – 45) in relation to the material that follows. Additionally, expanding on the novelty of the Ag₁₆S₄ inner core structure (lines 141- 144) would be welcomed.

Also, the author's do not mention if the as-synthesized clusters were "washed" or re-crystallized after isolation to remove excess impurities prior to subsequent analyses. This can be helpful for obtaining higher quality single crystals for XRD.

Title: “Small Symmetry-Breaking Triggering Large Chiroptical Responses of Ag₇₀ Nanoclusters”

Author(s): Xi-Ming Luo, Chun-Hua Gong, Fangfang Pan, Yubing Si, Jia-Wang Yuan, Muhammad Asad, Xi-Yan Dong, Shuang-Quan Zang, and Thomas C. W. Mak

Reviewer(s)' Comments:

Reviewer 1:

This is an excellent and well-written paper on a new Ag₇₀ nanocluster." *It deserves serious consideration for publication in Nature Communications.* However, in its present form, the manuscript has several “deficiencies” that need to be ratified:

Response: We appreciate **Reviewer 1**'s positive comments very much. The useful suggestions help to improve our manuscript. In the revised manuscript, we addressed all concerns of **Reviewer 1**.

1). The formulation of the title cluster is {Ag₇₀S₄(S-iPr)₂₄(TFA)₂₀(DMF)₃}²⁻ (Rac-Ag₇₀). Here CF₃COO⁻ is abbreviated as TFA. What is the formulation of companion cluster Ag₁₂ in the cocrystal Ag₇₀·Ag₁₂? It looks like this small Ag₁₂ cluster is a highly distorted cuboctahedron (Fig. S43). Yet the large Ag₇₀ cluster, under the influence of this small companion Ag₁₂ cluster, has “reverted” back to achiral of T_d symmetry! With TFA ligands, Ag₇₀ crystalizes as a racemate Rac-Ag₇₀ with mirror-images of R and S chiral clusters: how did this small, distorted Ag₁₂ cluster, help make the large Ag₇₀ cluster lose its chirality?

Response: As suggested by **Reviewer 1**, the molecular formula of Ag₁₂ in the cocrystal Ag₇₀·Ag₁₂ is determined to be Ag₁₂(SⁱPr)₆(CF₃COO)₆ by SCXRD analysis (Supplementary Fig. 46) and ESI-MS (Supplementary Fig. 54 and Supplementary Table 10).

We think that the lower T-symmetry of chiral clusters is induced by the slight lattice distortion, according to single-crystal structural analysis (Supplementary Figs. 46–50). When the small Ag₁₂ cluster occupies the interstitial space, it probably weakens the strain in individual chiral cluster in Rac-Ag₇₀ crystals, resulting in a rise of symmetry to higher T_d-symmetry.

2). Please provide structural details of the Ag₁₂ cluster, complete with the ligand positions and interatomic distances, in the cocrystal Ag₇₀·Ag₁₂. Also provide the UV-Vis and CD of the Ag₁₂ cluster in pristine form if possible. If this Ag₁₂ cluster cannot be made, I would like to suggest DFT calculations to identify its contribution to the UV-Vis and CD spectra of the cocrystal, if any.

Response: As suggested by **Reviewer 1**, the structural details of the Ag₁₂ cluster together with the ligand positions and interatomic distances in the cocrystal **Ag₇₀·Ag₁₂** have been added in the revised manuscript.

“The smaller Ag(I)₁₂ cluster possesses a slightly distorted cuboctahedral metal framework (Ag⋯Ag distances: 3.06(1)–3.09(2) Å, Supplementary Figs. 46–48). The six μ_4 -SⁱPr[−] ligands are anchored on the quadrilateral surface of Ag(I)₁₂ with Ag–S bond lengths in the range of 2.44(4)–2.50(4) Å, and the CF₃COO[−] ligands on the edges in μ_2 - η^1, η^1 coordination mode (Ag–O bond lengths: 2.29(2)–2.44(4) Å).”

As suggested by Reviewer 1, we identify the contribution to the UV-Vis spectrum of Ag₁₂ in the cocrystal **Ag₇₀·Ag₁₂** through DFT calculations (at the PBE0/def2SVP level, Fig. R1), because we indeed cannot isolate the identical Ag₁₂ after a lot of trials. The theoretical results show that the Ag₁₂ cluster has not contribution to the absorbance in the visible-light region, implying that the CD signals in the ligand-exchange process come from the superatomic Ag₇₀.

Figure R1. Experimental (**Ag₇₀·Ag₁₂** and **Rac-Ag₇₀**) and calculated (**Ag₇₀** and **Ag₁₂**) absorption spectra.

3). As the TFA is replaced by the chiral TFL ligands, the CD of the Ag₇₀ is progressively enhanced, indicating deracemization. Is this experimental observation the result of (1) the instant conversion of, say, S to R enantiomer, or (2) a progressive distortion of the metal framework as more and more TFAs are replaced by the chiral TFLs? If the answer is (1), how many TFAs must be replaced by chiral TFLs to cause this transformation?

Response: This is a very good comment. We think that the two processes (1) and (2) could cooccur as the TFA is replaced by the chiral TFL ligands.

First, for process (1), we could find implications in the back CD titration of deracemization solution by using achiral TFA. When n(CF₃COOAg):n(R-TFLAg) reaches 1000:50, the CD signal nearly approaches zero and there is basically no change afterwards. ESI-MS results show that the chiral TFL nearly disappears (Supplementary Figs. 36–37 and Supplementary Tables 8–9). In addition, the successful crystallization of chiral crystals with mirror-image CD response using chiral TFL also suggests that

the conversion of chirality really occurs in this system (Fig. 5c). Nevertheless, because of the complex dynamic process in the solution system, where each cluster could have a different occurrence for the number of chiral ligands, the conversion point could not be completely quantified for the time being. In future work, we will work on this interesting topic.

Second, for process (2), the progressive distortion of the Ag₇₀ framework also can be deduced, because the CD intensity rises progressively and reaches a peak, after which CD profiles keep constant, implying that the distortion has a limit. The more serious distortions of Ag₇₀ skeleton are actually found in the monochiral single-crystals (Fig. 5a–b). In addition, after several iterative crystallization using TFL ligands, we fortunately obtain a higher-quality single crystal for diffraction, whose cif (**S-Ag₇₀-3.cif**) is uploaded for Reviewers and whose CD spectra (Fig. R2) agree well with the original data, supporting the distortion become more and more with the increasing of chiral TFLs.

Figure R2. CD spectra of **R/S-Ag₇₀** and **S-Ag₇₀-3** crystallites.

4). The UV-Vis remains unchanged throughout the TFA-TFL exchange reaction. More importantly, this exchange reaction should be followed/tracked (in sync) by CD, FTIR, NMR to determine the ratio of TFL:TFA ligands on the surface. This ratio may or may not (my guess is NOT) correspond to the experimental ratio used. The purpose of this tracking is to determine the *m* value in $\{\text{Ag}_{70}\text{S}_4(\text{S}^i\text{Pr})_{24}(\text{TFL})_{20-m}(\text{TFA})_m(\text{DMF})_3\}^{2-}$ and to correlate it with the CD signal.

Response: As suggested by **Reviewer 1**, the exchange reaction is in sync tracked by CD (Fig.4 and Supplementary Figs. 31–32) and ¹⁹F-NMR spectra (Fig. R3 and Table R1).

As commented by **Reviewer 1**, the ratio on the interface of the actual cluster does not match the experimental ratio used, which is observed in the ESI-MS data (Supplementary Figs. 35–37).

In the range of $n(\text{R-/S-TFLAg}):n(\text{Ag}_{70}) = 0\text{--}15$ (the experimental ratio used), the CD intensity is almost linearly enhanced (Fig. 4c and Supplementary Fig. 32c), which should include two basic inducements, e.g. chiral conversion of a part of clusters and chiral enlargement of another part of clusters. Therefore, in combination ESI-MS data, we proposed that m value should be an average in some range.

For in sync ^{19}F -NMR spectra of exchange reaction using a chiral metal precursor (Fig. R3), we found that with different amounts of precursors (R-TFLAg) added, there are always only two peak positions, which are different from those in precursor of TFAAg and TFLAg (Supplementary Fig. 63), indicating the different shielding effects of the Ag_{70} skeleton on F nuclei of TFA and TFL. ^{19}F -NMR spectra show that during exchange reaction, the ratios of TFA to TFL are close to the feed ratio (Table R1).

Figure R3. ^{19}F -NMR spectra of **Rac-Ag₇₀** using a chiral metal precursor (R-TFLAg) in DMF and CDCl_3 at room temperature.

Table R1. The ratio of $n(\text{TFA})$ to $n(\text{TFL})$ calculated by ^{19}F -NMR data in the progress of **Rac-Ag₇₀** with different amounts of chiral metal precursor (R-TFLAg) in DMF and CDCl_3 at room temperature.

$n(\text{Ag}_{70}):n(\text{R-TFLAg})^{\text{a}}$	$n(\text{TFA}):n(\text{TFL})^{\text{b}}$	
	Exp.	Cal.
1:0	1:0	1:0
1:5	1:0.23	1:0.2
1:10	1:0.44	1:0.33
1:20	1:1.2	1:1
1:50	1:2.5	1:2.5

^a The experimental ratio of Ag_{70} cluster to the added R-TFLAg. ^b The ratio of TFA to TFL calculated by ^{19}F -NMR. ^c The experimental ratio of TFA of the parental Ag_{70}

cluster to the added R-TFL of R-TFLAg.

Supplementary Figure 63. ¹⁹F-NMR spectra of **R-Ag₇₀**, **S-Ag₇₀**, **Rac-Ag₇₀**, TFLAg, and CF₃COOAg in DMF and CDCl₃ at room temperature.

“The ¹⁹F-NMR spectrum of **Rac-Ag₇₀** shows one resonance (−74.2 ppm), which corresponds to the −CF₃ group in the CF₃COO[−] ligand, while the resonance shift to −73.3 ppm for CF₃COOAg in DMF and CDCl₃. The ¹⁹F-NMR spectra of **R-Ag₇₀** and **S-Ag₇₀** show two resonances (−74.5 and −75.8 ppm) in *ca.* 12:8 ratio, corresponding to the −CF₃ groups in the CF₃COO[−] and TFL[−] ligands, respectively. For comparison, the resonance corresponding to the −CF₃ groups in the TFL[−] ligand shifts to −76.2 ppm for TFLAg in CDCl₃.”

5). Fig. 32c provides ESI-MS of {Ag₇₀S₄(S^{*i*}Pr)₂₄(TFL)_{20−*m*}(TFA)_{*m*}(DMF)₃}^{2−}·*x*(TFL)Ag. What is the role of the extra (TFL)Ag complexes? Are they surface attachments?

Response: Reviewer 1's comments have deeply enlightening significance.

As commented by **Reviewer 1**, the additional (TFL)Ag complexes are found to be a commensal with Ag₇₀ in ESI-MS of {Ag₇₀S₄(S^{*i*}Pr)₂₄(TFL)_{20−*m*}(TFA)_{*m*}(DMF)₃}^{2−}·*x*(TFL)Ag. The similar observations are subject to ESI-MS of **Rac-Ag₇₀** (Supplementary Fig. 29), **Rac-Ag₇₀** + CF₃COOAg (Supplementary Fig. 29), **Rac-Ag₇₀** + C₂F₅COOAg (Supplementary Fig. 30), and **Ag₇₀·Ag₁₂** (Supplementary Fig. 54). We think that there could be potential sites on the surface of Ag₇₀ clusters, which have more affinity with the small molecules including (TFL)Ag, CF₃COOAg and C₂F₅COOAg.

In fact, we also pay attention to this interesting point and are working on it to assemble such large superatomic clusters by virtue of these potential sites.

6). This paper provides no structural details on the surface ligands for the STAR cluster: the chiral R-Ag₇₀. What is the TFL:TFA ratio on the surface? Are they badly

disordered? Perhaps FTIR or NMR can provide an estimate (see Item 4 above)

Response: As commented by **Reviewer 1**, the surface ligands of cluster surface are badly disordered, resulting in the loss of structural details.

As suggested by **Reviewer 1**, we measured ^{19}F -NMR of **R-Ag₇₀** samples in CDCl_3 and found two resonances at -74.5 and -75.8 ppm in *ca.* 12:8 ratio (**Supplementary Fig. 63**), corresponding to the $-\text{CF}_3$ groups in TFA and TFL ligands. So, the TFA:TFL ratio on the surface is *ca.* 12:8 by ^{19}F -NMR.

“The ^{19}F -NMR spectrum of **Rac-Ag₇₀** shows one resonance (-74.2 ppm), which corresponds to the $-\text{CF}_3$ group in the CF_3COO^- ligand, while the resonance shift to -73.3 ppm for CF_3COOAg in DMF and CDCl_3 . The ^{19}F -NMR spectra of **R-Ag₇₀** and **S-Ag₇₀** show two resonances (-74.5 and -75.8 ppm) in *ca.* 12:8 ratio, corresponding to the $-\text{CF}_3$ groups in the CF_3COO^- and TFL^- ligands, respectively. For comparison, the resonance corresponding to the $-\text{CF}_3$ groups in the TFL^- ligand shifts to -76.2 ppm for **TFLAg** in CDCl_3 .”

7). It was stated in the paper that “Only one case of enantiomerically pure single crystals of a medium-sized silver nanocluster, Ag_{78} , protected by a chiral diphosphine ligand, has been characterized.²¹” This is utterly not true. The following are just a few prior examples: *Angew. Chem. Int. Ed.*, 2021, DOI: 10.1002/anie.202108141; *Angew. Chem. Int. Ed.*, 2021, 60(23): 12897–12903; *J. Am. Chem. Soc.*, 2021, 143(27): 10214–10220; *Angew. Chem. Int. Ed.* 2018, 57(13): 3421–3425.

Response: We define a medium-sized silver nanocluster with the number of Ag atoms over 50 (*J. Am. Chem. Soc.* 2019, 141, 19754–19764). Therefore, we claim that Ag_{78} reported by Zheng group is the most outstanding one. In addition, these significant small-sized Ag clusters (< 50 Ag) recommended by **Reviewer 1** are also referenced (Ref 5, 6, 8, 10).

Reviewer 2:

Crystallography Report

This took me a while because I am entering unfamiliar territory here. For this study, the question is not: Are these fully refined crystal structures? They clearly aren't, and don't claim to be. The question is rather: Do the data support the conclusions the authors draw from them? My tentative answer to this is: As far as I can tell they do. I have my doubts about the charges of the clusters, simply due to the use of SQUEEZE, but from what I can tell the main conclusions about the chiral shape of the clusters are not affected by the number of valence electrons within it (and I do not understand the spectroscopic evidence, so whether that supports the charges of the clusters other referees have to answer).

I have a slight reservation about the P21 structure: I agree with the authors that the most likely space group is indeed P21; I have tried to solve the structure in P21/c and

got nowhere, and ADDSYMM from PLATON did not suggest any higher symmetry. However, as the authors state correctly the FLACK parameter is very high, and this can sometimes resolve itself upon full refinement of the structure. What I have also sometimes observed is that distortions in molecules occur together with high FLACK parameters in early refinement stages which both disappear when the structure is fully resolved. I cannot rule out from the data presented here that this has not occurred here, i.e. that the small differences between the AG70 cluster in P-3c1 and in P21 are due to incomplete refinement. Again, I do not understand the spectroscopic data about the shape of the clusters, so if that indeed shows that the additional distortion is real then my comment above is void.

Response: Thank this reviewer for the modest comments very much.

The severe disorder and some other ambiguous facts in the crystal data indeed plagued us a lot. Although as the reviewer stating “I cannot rule out from the data presented here that this has not occurred here”, there is also possibility that the structure should be in centrosymmetric space group, which we admit, is actually a more common selection, particularly for small molecule crystallography. Luckily, after hundreds of crystallization experiments, we obtained a new crystalline phase for the same cluster that allowed us to unambiguously determine the space group as P2₁2₁2₁(S-**Ag70-3.cif**). The single-crystal X-ray diffraction experiments showed that the structure is in an orthorhombic system and the unit cell parameters (23.89, 29.99, 46.30, 90, 90, 90) are completely different from the one in the previous manuscript (23.83, 33.56, 37.56, 90, 91.23, 90) and more closer to the parent structure of **Rac-Ag70** (29.64, 29.64, 46.99, 90, 90, 120). Although only the Ag–S skeleton (the same Ag₇₀S₂₈) can be well solved in the new structure, the statistic diffraction intensity gives a lower $|E^*E-1|$ value of 0.915, and only chiral space groups are suggested according to the systematic absence. Finally, after solving the Ag₇₀S₂₈ core, a much lower Hooft y parameter (0.158(12))^{Ref 1} was computed from the least-squares refinement on F², implying the overwhelming probability of the R-chirality of the cluster in the crystal. Note that Hooft y parameter has been widely used in present to determine the absolute configuration in modern crystallography. It is regarded as giving even greater enantiomer distinguishing power than Flack parameter.

In addition, we carefully compared the Ag₇₀ skeleton (**Fig. R4**) in the new structure with that in the structure of **Rac-Ag70**. Clear distortion of the new cluster can be observed, which also supports its non-centrosymmetric geometry induced by the peripheral ligands. The new crystal data (S-**Ag70-3.cif**) can better support our conclusions in the manuscript, which is uploaded for the Reviewer’s evaluation.

Figure R4. Compared the Ag₇₀ skeleton in the new structure (blue, from S-Ag₇₀-3.cif) with that in the structure of **Rac-Ag₇₀** (green).

The charge state of Ag₇₀ of **Rac-Ag₇₀**, **Ag₇₀·Ag₁₂** and **R/S-Ag₇₀** is determined to be -2 , based on the jellium model, and supported by ESI-MS data (Supplementary Figs. 29, 35–37, 54; and Supplementary Tables 2–10).

Ref 1: Hooft R. B. W., Straver L. H., Spek A. L. Determination of absolute structure using Bayesian statistics on Bijvoet differences. *J. Applied Cryst.* (2008) 41, 96–103.

Reviewer 3:

The article can be published after the following questions are addressed.

1). In the structural description, only the metal framework was discussed. The coordination of thiolates, trifluoroacetates, and DMF ligands should be added. Are there any changes of Ag...Ag, Ag–S, and Ag–O bonds between T- and T_d-symmetric Ag₇₀? The bond lengths or angles should be reported, which are the benchmark to identify whether the structure is distorted.

Response: As suggested by **Reviewer 3**, the coordination of thiolates, trifluoroacetates, DMF ligands, and relevant Ag...Ag, Ag–S, and Ag–O bonds had been discussed in the revised manuscript and revised Supplementary Information.

In addition, the relevant differences of Ag...Ag, Ag–S bonds between T- and T_d-symmetric Ag₇₀ is included in Supplementary Table 1.

2). The crystals of Rac-Ag₇₀ and Ag₇₀·Ag₁₂ were obtained from the same synthetic procedure (in experimental section, page 14). How did the authors said that “By using a small Ag(I) cluster as a stabilizer,...”? Based on the description of the experimental

section, those two crystals can be obtained in the same condition. How to separate the two in the same reaction since the colors of the crystals are similar.

Response: It is likely that we did not clearly describe the difference of synthesis conditions between Rac-Ag_{70} and $\text{Ag}_{70}\cdot\text{Ag}_{12}$, leading to **Reviewer 3's** misunderstanding.

The synthetic procedure of the crystals of Rac-Ag_{70} and $\text{Ag}_{70}\cdot\text{Ag}_{12}$ are similar, but the solvent is different. For Rac-Ag_{70} : 3.5 mL $i\text{PrOH}$ and 1.5 mL DMF; For $\text{Ag}_{70}\cdot\text{Ag}_{12}$: 3.5 mL EtOH and 1.5 mL DMF. We use different alcohols to tune the reducibility of DMF in reaction systems. Therefore, pure Rac-Ag_{70} and $\text{Ag}_{70}\cdot\text{Ag}_{12}$ crystals are separately prepared in respective systems.

In order to avoid the readers' misunderstanding, we revise the related description in Methods-Synthesis, which is highlighted in yellow in the revised manuscript.

3). The metal framework possesses T_d symmetry in $\text{Ag}_{70}\cdot\text{Ag}_{12}$. Will it be the same symmetry if it contains the surficial ligands? In page 12, the author mentioned that "the small Ag_{12} cluster not only stabilizes Ag_{70} but also changes the crystal packing, which probably weakens the intercluster strain in Rac-Ag_{70} and result in T_d -symmetric Ag_{70} in $\text{Ag}_{70}\cdot\text{Ag}_{12}$." Are there any interactions such as $\text{H}\cdots\text{F}$ contacts between clusters in Rac-Ag_{70} ? If so, please consider them for discussion.

Response: The 70-nuclearity Ag NC including metal framework and its surficial ligands in $\text{Ag}_{70}\cdot\text{Ag}_{12}$ exhibits nearly perfect T_d symmetry.

At the same time, the $\text{C-H}\cdots\text{F}$ and $\text{C-H}\cdots\text{O}$ interactions between clusters in Rac-Ag_{70} have been discussed in detail in the revised manuscript. It is found that there are multiple $\text{C-H}\cdots\text{F}$ hydrogen bonds between CF_3COO^- and $-\text{CH}_3$ groups of S^iPr^- and DMF ligands (**Supplementary Fig. 16**), suggesting the strong interactions between the clusters.

Supplementary Figure 16. The hydrogen bonds of Rac-Ag_{70} . (a) The intra-cluster and

inter-cluster C–H···F, and intra-cluster C–H···O hydrogen bonds. (b) The inter-cluster C–H···F hydrogen bonds. (c) The intra-cluster C–H···F, and C–H···O hydrogen bonds.

4). It is clever to add chiral ligands (TFL⁻) to get optically pure R-/S-Ag₇₀. R-Ag₇₀ crystallized in a non-centrosymmetric space group $P2_1$, but the flack parameter (0.42) shows large uncertainty of the determination of the absolute structure. It could be due to racemic or twinned. However, the CD spectrum shows optical activity nicely. Thus this reviewer believes the R-/S-Ag₇₀ were synthesized. The Figure 4d is quite unclear while R-/S-TFLAg and Rac-Ag₇₀+R/S-TFLAg are stacked together. Please revise.

Response: We appreciate **Reviewer 3**'s positive comments on our conclusions.

As **Reviewer 3** suggested, **Fig. 4d** has been revised in the revised manuscript. And the control CD of R/S-TFLAg and deracemization of **Rac-Ag₇₀** solution have been shown in **Supplementary Fig. 34**.

5). Line 352, Page 15, “no-H” should be “ non-hydrogen”.

Line 180, Page 9, “m/z” should be “*m/z*” (in italic style)

Response: As **Reviewer 3** suggested, the errors mentioned have been corrected in the revised manuscript.

Reviewer 4:

This paper presents the structure and optical properties of chiral Ag₇₀ nanoclusters. A crystal structure of racemic Ag₇₀ nanoclusters is first presented. This material was then de-racemized by incorporating a chiral silver precursor into the synthesis, and the crystals of resulting R/S enantiomeric pairs were also solved. De-racemization was also monitored using CD and was found to be reversible. In particular, deracemization of such clusters for R/S structure solutions is of particular significance, as it remains a challenge. The charge state of rac-Ag₇₀ is determined to be -2, based on the jellium model, and supported by ESI-MS data. The UV-Vis-IR spectra also conform to that expected of a molecular cluster, and are supported by theoretical simulations. The structural analysis provided suggests the surface Ag-S moieties cause a minor Td symmetry breaking distortion, resulting in two enantiomers with large chiroptical properties. The CD data supports this. However, the author's note that the ligand layer structure was unable to be solved with single crystal XRD, which is not entirely unexpected given the fragility of similar metal nanoclusters when exposed to X-rays. A co-crystal of Ag₇₀-Ag₁₂ was also solved, and found to impart greater stability than the Rac-Ag₇₀ in solution. A stability mechanism of Ag₇₀ clusters in solution is proposed. The paper also presents a novel Ag₁₆S₄ FCC inner core structure within the Ag₇₀ cluster, *which is particularly of interest to the community. On the whole, the data in the paper is convincing and well presented, and should be published for the community. I do not have major remarks other than the following:*

1). A few parts of the main text are unclear, which could undermine the importance of this work. In particular, the introductory sentences are somewhat clunky/confusing

(lines 40 – 45) in relation to the material that follows. Additionally, expanding on the novelty of the Ag_{16}S_4 inner core structure (lines 141- 144) would be welcomed.

Response: We really appreciate the constructive comment.

As **Reviewer 4** suggested, the introduction section (lines 40 – 45) has been rewritten which is highlighted in yellow in the revised manuscript.

The novelty of Ag_{16}S_4 inner core structure has been highlighted and expanded in the revised manuscript (Page 7).

2). Also, the author's do not mention if the as-synthesized clusters were “washed” or re-crystallized after isolation to remove excess impurities prior to subsequent analyses. This can be helpful for obtaining higher quality single crystals for XRD.

Response: Thanks for your suggestion very much. We apologize for not describing the post-processing method of these crystals in detail.

Pure and clean crystals of the as-synthesized clusters crystallized in the vessel when the reaction is finished. So we simply washed these crystals with dichloride/n-hexane. The revised description (Methods-Synthesis section) is highlighted in yellow in the revised manuscript.

REVIEWER COMMENTS

Reviewer #1 (Remarks to the Author):

The authors have done a great job in revising the paper. Referring to my original Comments 3-5, the authors have tracked the TFA-TFL exchange reaction by CD, FTIR, ESI-MS, and ¹⁹F-NMR to determine the ratio of TFL:TFA ligands on the surface. The ultimate/stable composition is {Ag₇₀S₄(S-iPr)₂₄(TFL)₈(TFA)₁₂(DMF)₃}²⁻ (composition A) or {Ag₇₀S₄(S-iPr)₂₄(TFL)₈(TFA)₁₂(DMF)₃}²⁻ (1-4) (TFL)Ag (composition B).

This conclusion is important; it should be summarized in the text and the experimental evidence should be provided in the SI (not just in the Rebuttal). The authors should also clarify the composition of the ultimate/stable cluster in solution is A or B, as judged by ESI-MS and ¹⁹F-NMR spectra. Furthermore, what are the roles of TFL (chiral organic ligand) and (TFL)Ag (chiral complex) in the deracemization process?

In addition, the following errors should be corrected:

1. Fig. 1b and Fig. 5a: “-4C₃” is incorrect, as 4C₃ is maintained in the chiral cluster of T symmetry (only the loss of 4σ_d reflection planes)!
2. Line 317: the statement “...seriously distorted...” should be changed to “...severely distorted...”

Reviewer #2 (Remarks to the Author):

Crystallography Report

As I did not have any suggestions for improvements, merely comments on the interpretation of the data, I am quite happy to see the paper published, especially as I can see that all things I was unsure about there addressed by responses to other referees.

Reviewer #3 (Remarks to the Author):

The authors have addressed most of the queries and the paper can be published in its present form.

Thank Reviewers for the very useful suggestions.

Reviewer(s)' Comments:

Reviewer 1 (Remarks to the Author):

The authors have done a great job in revising the paper.

Response: We appreciate **Reviewer 1**'s positive comments very much.

Referring to my original **Comments 3-5**, the authors have tracked the TFA-TFL exchange reaction by CD, FTIR, ESI-MS, and ¹⁹F-NMR to determine the ratio of TFL:TFA ligands on the surface. The ultimate/stable composition is {Ag₇₀S₄(SⁱPr)₂₄(TFL)₈(TFA)₁₂(DMF)₃}²⁻ (composition A) or {Ag₇₀S₄(SⁱPr)₂₄(TFL)₈(TFA)₁₂(DMF)₃}²⁻⁽¹⁻⁴⁾(TFL)Ag (composition B).

This conclusion is important; it should be summarized in the text and the experimental evidence should be provided in the SI (not just in the Rebuttal).

Response: This is a very good suggestion. The conclusion is summarized in the revised Manuscript (Pages 11–12 and Page 14) and the experimental evidences are provided in the Supplementary Information (Supplementary Fig. 39 and Supplementary Fig. 59).

The authors should also clarify the composition of the ultimate/stable cluster in solution is A or B, as judged by ESI-MS and ¹⁹F-NMR spectra.

Furthermore, what are the roles of TFL⁻ (chiral organic ligand) and (TFL)Ag (chiral complex) in the deracemization process?

Response: As suggested by **Reviewer 1**, this point has been clarified in the revised Manuscript (Page 11). ESI-MS (Supplementary Figs. 35–37 and Supplementary Tables 6–9) demonstrates that A and B coexist in the solution of the exchange reaction. By crystallization, only the composition A is generated in crystalline phase.

Combining the single-crystal structure analysis, and the nearly identical CD signals of solution (containing A and B) and solid state (merely containing A), we propose that the incoming TFL⁻ (chiral organic ligand) induces the structural deformation, which plays a critical role in chiroptical response and the deracemization process. In addition, (TFL)Ag (chiral complex), which was used to trigger the reaction in our experiment, is found to be the concomitants of the silver cluster in ESI-MS (Supplementary Figs. 35–37 and Supplementary Tables 6–9), and also plays a role in stabilizing the clusters (Supplementary Fig. 31d).

In addition, the following errors should be corrected:

1. Fig. 1b and Fig. 5a: “-4C₃” is incorrect, as 4C₃ is maintained in the chiral cluster

of T symmetry (only the loss of $4\sigma_d$ reflection planes)!

Response: Thank **Reviewer 1** for the nice reminder very much. As **Reviewer 1** comments, the T symmetry has the C_3 axis.

Nevertheless, for the chiral clusters in our work, they lack the C_3 axis and do not belong to the T symmetry, which should be strictly called C_1 symmetry (Supplementary Figs. 61–62). Because the degree of symmetry-breaking of the chiral cluster is small relative to the T symmetry, we called it pseudo- T -symmetry. These explanations are included in the revised manuscript (Page 3 and Page 14).

2. Line 317: the statement “...seriously distorted...” should be changed to “...severely distorted...”

Response: As **Reviewer 1** suggested, the error mentioned has been corrected in the revised manuscript.

Reviewer 2 (Remarks to the Author):

Crystallography Report

As I did not have any suggestions for improvements, merely comments on the interpretation of the data, I am quite happy to see the paper published, especially as I can see that all things I was unsure about there addressed by responses to other referees.

Response: We sincerely thank **Reviewer 2** for the positive comments.

Reviewer 3 (Remarks to the Author):

The authors have addressed most of the queries and the paper can be published in its present form.

Response: We appreciate **Reviewer 3** for the recommendation very much.

REVIEWER COMMENTS

Reviewer #1 (Remarks to the Author):

In my opinion, this manuscript is now acceptable for publication.

Thank Reviewers for the very useful suggestions.

Reviewer(s)' Comments:

Reviewer 1 (Remarks to the Author):

In my opinion, this manuscript is now acceptable for publication.

Response: We sincerely thank **Reviewer 1** for the positive comments.